# An Examination of the Number of Adolescent Scoliotic Curves That Are Braceable at First Presentation to a Scoliosis Service

**DOI:** 10.3390/healthcare11030445

**Published:** 2023-02-03

**Authors:** Laura Hartley, Conor Jones, Darren Lui, Jason Bernard, Timothy Bishop, Jan Herzog, Daniel Chan, Oliver Stokes, Adrian Gardner

**Affiliations:** 1The Royal Orthopaedic Hospital, NHS Foundation Trust, Birmingham B31 2AP, UK; 2The Royal Devon and Exeter, NHS Foundation Trust, Exeter EX2 5DW, UK; 3St George’s University Hospitals, NHS Foundation Trust, London SW17 0OT, UK

**Keywords:** AIS, adolescent, scoliosis, bracing

## Abstract

Adolescent idiopathic scoliosis (AIS) affects between 0.5% and 5.2% of adolescents and is progressive in two-thirds of cases. Bracing is an effective non-operative treatment for AIS and has been shown to prevent up to 72% of curves from requiring surgery. This paper explores the presentation of AIS in the UK and identifies who would be suitable for bracing, as per guidelines published by the Scoliosis Research Society (SRS) and British Scoliosis Society (BSS), through curve severity and skeletal maturity at presentation. There were 526 patients with AIS eligible for inclusion across three tertiary referral centres in the UK. The study period was individualised to each centre, between January 2012 and December 2021. Only 10% were appropriate for bracing via either SRS or BSS criteria. The rest were either too old, skeletally mature or had a curve size too large to benefit. By the end of data collection, 38% had undergone surgery for their scoliosis. In the UK, bracing for AIS is only suitable for a small number at presentation. Future efforts to minimise delays in specialist review and intervention will increase the number of those with AIS suitable for bracing and reduce the number and burden of operative interventions for AIS in the UK.

## 1. Introduction

Adolescent idiopathic scoliosis (AIS) is a three-dimensional deformity of the spine of unknown aetiology, seen in individuals aged 10–18 years and characterised by a coronal curvature greater than 10° with vertebral rotation [1]. The condition affects between 0.5% and 5.2% of adolescents and is progressive in two-thirds of cases [2].

Surgical intervention is indicated in those with large curves, commonly agreed to be greater than 50°, measured using the Cobb angle [3], amongst other criteria that include vertebral body rotation and sagittal balance. Although effective at changing spinal and thoracic shape [4], surgery leads to stiffness of the spine, caused by the instrumented fusion, and has potential medical, psychological and social implications for the children and their families [5].

The natural course of AIS is that small, flexible, single curves can progress to larger, stiffer and often double curves as children progress through the adolescent growth spurt [6]. When operated on, these curves are then associated with an increase in the required magnitude of surgery, with a concomitant increase in the potential for complications and morbidity. The early identification of and intervention with smaller curves has been shown to reduce the number of curves that progress and require more significant intervention, including surgery [7]. In some areas of the world, earlier identification of scoliosis is sought via both screening and education programmes [8] in the community, but the utility of scoliosis screening has been questioned in a number of countries [9,10,11]. As a result, neither the United Kingdom National Screening Committee (UKNSC) nor the United States Preventative Services Task Force (USPSTF) currently recommend scoliosis screening programmes for AIS [10,12].

Thoracolumbosacral orthoses (TLSOs) have been shown to be an effective non-operative treatment for AIS [13]. The Bracing in Adolescent Idiopathic Scoliosis Trial (BRAIST) demonstrated that 72% of braced individuals were prevented from subsequently developing curves that would require surgical intervention [13]. As such, the use of a TLSO is recommended by the Scoliosis Research Society (SRS) for the management of AIS in skeletally immature individuals with mild to moderate curves (defined as either Risser grade 0–1 with a coronal Cobb angle of 20–40°, or Risser grade 2–3 with a coronal Cobb angle of 30–40°) [13,14]. More recently, the British Scoliosis Society (BSS) [15] has clarified these guidelines, to state that bracing should be considered the primary method of management for AIS in adolescents that are aged between 10 and 15 years with a curve that measures between 20° and 40° degrees and with the potential for future growth, as defined by a Risser grade of 0–2, along with a curve apex seen at T7 or lower. The BSS guidelines [15] are the basis on which the BASIS randomised control trial [16], which compares full-time to night-time bracing for AIS, is currently recruiting within the UK.

This study aims to identify the number of adolescents at first referral to a UK scoliosis centre that would fulfil the criteria of being suitable for bracing as the initial method of management, following either the SRS or BSS guidelines [14,15].

## 2. Methods

This study was conducted across three tertiary referral centres for spinal deformity in the UK. The first centre, which serves a catchment area of approximately 2 million people in the south-west of England, with a predominantly rural, white British population, reviewed records for all individuals referred between January 2012 and December 2016. The second centre, which serves approximately 1.3 million people within Greater London and the south-east of England, with a predominantly urban population, where approximately half of all residents identify as white British, reviewed patients referred between January 2015 and December 2016. The third centre, which serves a population of 6 million across Birmingham and the West Midlands, with a mostly urban population of varied ethnicities, reviewed all referrals between January 2015 and December 2021. The different time periods for the three centres represent the time periods that data were accessible for review under the terms of the IRB permissions granted from each site, and represent a convenience sampling technique. All consecutive case referrals during those time periods were identified and reviewed.

All cases of scoliosis were identified retrospectively, from prospectively compiled outpatient clinic booking databases, along with business information services at each individual centre, with subsequent reviews of both clinical records and radiographs. For clarity, referrals to the three scoliosis units were made in the absence of a specific AIS screening or public education programme. All scoliotic curves in the referred individuals were initially identified by primary care physicians or other healthcare professionals within the community. The inclusion criteria for an individual to be part of this review were that the individual was aged between 10 and 18 years inclusive, with a clinical diagnosis of AIS. Those with spinal curvatures secondary to other aetiologies, an age under 10 years or over 18 years at referral (and thus not defined as AIS) [17], along with those initially investigated or treated at other scoliosis centres prior to being referred to one of the three centres, were excluded in this study.

The study was conducted with local institutional board approval (IRB reference 21-064). Global ethical approval and individual consent of study participants were deemed not to be required following the guidance of the UK Health Research Authority, since this was an anonymised, retrospective review of practice.

The radiographic measures recorded were the size of the scoliotic curvature on the initial assessment radiograph, measured using the coronal Cobb angle [3], along with the Risser sign [18,19] as a measure of skeletal maturity, at the time of first presentation to the tertiary centre using PACS Insignia imaging software tools [20]. The subsequent management undertaken in each case was also recorded.

The SRS and BSS bracing criteria [14,15] were used to identify those individuals that would have been considered eligible for bracing. Statistical analysis was performed using R statistical environment 3.0.2 [21]. Data are presented as the median, interquartile range or range as appropriate. Statistical tests were performed, using *t* tests to compare normally distributed data (age) and tests of proportion to compare curve types between sexes, in addition to establishing whether there was a difference in the number of individuals deemed suitable for bracing between females and males.

## 3. Results

In total, there were 640 individuals referred to the tertiary centres with a diagnosis of scoliosis during the defined time periods. Of these, 526 cases met the inclusion criteria (Table 1). Non-idiopathic aetiologies of scoliosis excluded from the analysis included neuromuscular (*n* = 22), congenital (*n* = 21) and syndromic (*n* = 13) scoliosis. A total of 42 individuals with early-onset scoliosis (diagnosis made before the age of 10) were also excluded. There were 16 individuals who had been investigated and treated at other centres in the first instance who were also excluded. The demographics of this cohort are shown in Table 1.

Figure 1 shows the number of individuals for each Risser grade at presentation for both females (a) and males (b). Figure 2 shows the curve sizes stratified by Risser grade for females (a) and males (b), as box and whisker plots. Table 2 details the number of individuals, both female and male, that would have been suitable for bracing, as per the SRS or BSS criteria, based on the parameters measured at the initial presentation.

There is a difference between the number of individuals deemed suitable for bracing at initial presentation when the SRS [14] or the BSS [15] criteria for bracing are applied. However, when the totals from the two SRS groups are pooled, the application of the BSS criteria shows little difference from the total (*p* = 0.272). Notably, it is apparent that curve size and Risser grade do not relate well to each other. When using either the SRS [14] or BSS [15] criteria for a spinal curve size of 20–40°, there were 92 females and 27 males where the Risser grade was greater than 2. When using the SRS criteria [14] for a spinal curve size of 30–40°, there were 12 females and 1 male where the Risser grade was greater than 4. When using the BSS criteria [15] for age and curve size only, there were 44 females and 7 males with a Risser grade greater than 3, and 304 females and 50 males who were older than 15 years at presentation.

In total, 472 (90%) of those with AIS were beyond the bracing criteria recommended by the SRS [14], and 476 (90%) presented beyond the bracing criteria recommended by the BSS [15] at first presentation. Of the 232 females and 59 males that presented with an initial curve size of more than 40°, 170 females and 29 males (38% of the total cohort) had undergone surgical intervention at the end of the data collection periods.

## 4. Discussion

In the study reported here, only 10% (*n* = 54 by SRS criteria [14]; *n* = 50 by BSS criteria [15]) of individuals with AIS referred to three UK tertiary spinal deformity units met the criteria for bracing at the time of presentation. The majority of individuals referred presented with spinal curvatures too large and/or a skeletal maturity too advanced for successful bracing. Notably, the SRS [14] and BSS [15] criteria lead to a different number of individuals being deemed suitable for bracing (although not a statistically significant difference); however, in either case, both are a small number of the total. It is also worthy of note that, whilst this paper only looks at size of curve at first review, for both the SRS [14] and BSS [15] criteria, there will be a number of patients (that cannot be defined in this study) who present with a curve less than 20° in size and become braceable whilst under the care of a physician. Thus, the total number of braced patients may well be higher than reported here during the totality of treatment. As would be expected, the larger number of adolescents presenting with scoliosis are female; this reflects the known presentation of AIS, in general, as reported in the Bracing in Adolescent Idiopathic Scoliosis (BRAIST) study [13].

For individuals with mild to moderate curves, rigid bracing affords an elective, non-surgical means of preventing curve progression [13,14,15]. The BRAIST trial [13], which currently shows the best evidence regarding bracing in AIS, reported that 72% of braced individuals were prevented from developing curves that would subsequently need surgical intervention. To the best of our knowledge, this study is the first to assess curve severity at first presentation of those with AIS in the UK in the context of currently accepted bracing criteria. In a Norwegian study of 752 individuals, 39% presented with curve sizes greater than 40°, and 61% demonstrated skeletal maturity (Risser grade 3–5), making them ineligible for bracing at presentation [22]. Similar findings were reported in a Canadian study, in which 32% of confirmed AIS cases presented beyond the indications for bracing (either Risser grades 4–5 with a Cobb angle over 30° or any Risser grade with a Cobb angle over 40°) [23]. Other studies have also reported advanced curvatures in 22–56% of AIS presentations [24,25]. These studies, in conjunction with the data presented here, suggest that children are often referred later on in the natural course of the condition, such that a conservative management technique using bracing, as opposed to surgical intervention, is no longer appropriate. However, avoiding surgery is important, if possible and the BRAIST study has shown that bracing can do this [13]. Avoiding surgery avoids the downsides and risks of surgery, which, whilst statistically infrequent, include permanent neurological injury (up to and including paralysis) and blindness [26]. Additionally, as a curve size increases, operative treatment is associated with a longer operative time, a longer hospital stay and an increased risk of complications [27,28,29]. Advances in surgical techniques aim to reduce the risk of such adverse events; however, surgery will always carry a greater risk than conservative treatment [28,29,30,31,32].

There are multiple reasons why a referral to a tertiary specialist centre may be delayed. There are delays from the condition first being noticed to the presentation to a primary care physician. Further delays occur again between the primary care physician to the acceptance of the referral at the specialist centre, and then more time can elapse before the adolescent is first seen at the specialist centre. Moreover, there may first be a delay if scoliosis is not noticed by either the adolescent or their family or friends. Asymmetry of the posterior torso is seen in those without scoliosis as part of the range of normality in shape [4], and even if an asymmetry reflects the presence of underlying scoliosis, recognition depends on it being seen by others.

To be effective, bracing requires scoliosis to be identified and then reviewed by those that can instigate treatment promptly. It would be wrong to suggest that every child will be suitable for bracing, even if healthcare pathways were optimised and delays to treatment lifted. There are countries that employ school screening programmes in an attempt to circumvent these issues of recognition and delay. Such programmes are no longer supported by either the USPSTF or the UKNSC [10,11], who quote a lack of evidence to support the direct relationship between screening and improved quality-of-life outcomes. However, early detection is recommended in order to provide non-operative treatment options for AIS [33,34], and this depends on an early recognition of the condition. Scoliosis screening is advocated for by the Scoliosis Research Society, the Paediatric Orthopaedic Society of North America, the American Academy of Orthopaedic Surgeons and the American Academy of Pediatrics [35]. It is notable that school screening both increases the number of curves presented to a scoliosis clinic that are smaller in magnitude, and would thus benefit from bracing as a treatment, and also leads to a larger number of unnecessary evaluations for scoliosis in the worried well [36]. Whilst the work presented here is based in the UK, it is applicable for consideration in other healthcare systems around the world, where patients face similar delays in accessing specialist health services.

An issue that this work highlights is the discordance between the bracing guidelines and the measures of curve size and skeletal maturity. Assessing skeletal maturity is required to demonstrate that there is future potential growth in the adolescent, such that the use of a brace will lead to growth modulation of the spine. The ossification of the iliac apophysis describes the Risser grade [18], and historically this has been used as a measure of skeletal maturity. However, the relationship of the ossification of the iliac crest to peak height velocity is variable. While Risser grade 5 is a marker of adulthood, the understanding that Risser grades 0, 1 and 2 are before the adolescent growth spurt and that grades 3 and 4 are after the adolescent growth spurt is not correct [35]. It is recognised [35] that the Risser grade is a blunt tool for the assessment of future potential growth. It is possible that potential candidates for a successful outcome from bracing for AIS are being missed or discounted because of the Risser grade poorly reflecting peak height velocity and growth potential. As seen in this work, there were 119 children who presented with curves in the braceable range, but were too skeletally mature based on the BSS bracing criteria [15].

In an attempt to mitigate this issue, the ossification of other parts of the body has been analysed in relation to skeletal maturity and when the child attains peak height velocity. This includes the ossification of the humeral head [37], the hand and fingers (the Sanders clarification) [38]; the assessment of the ossification of the distal radius and ulna (DRU classification) [39]; the Sauvegrain method with ossification around the elbow [40]; and the ossification of the proximal femur [41], amongst others. The benefit of assessing the humeral head and proximal femur in this regard is that, like the iliac apophysis for the assessment of the Risser grade, the humeral head and proximal femur can be seen and assessed from the standard whole-spine radiograph used in clinical practice and, thus, no further radiographs are required. The Sanders score, Sauvegrain method and DRU scores require imaging of a part of the body not in the spine radiograph using a new exposure. That is associated with a small increase in cumulative radiation exposure, and thus risks the adolescent developing solid organ cancer in later life [42]. However, this has been circumvented to some extent by having the individual stand for their radiograph with one elbow flexed and the hand held out in the coronal plane at shoulder height [43]. This has been shown to be accurate and repeatable and within the imaging capabilities of low-dose imaging systems, such as the EOS system.

The benefits of the Sanders classification concern the detail of the system, with a larger number of scores compared to the Risser grade. Weaning from the brace should occur at a score of 8 [44]. A similar assessment has also been made using the DRU classification system [45]. Humeral head and proximal femur ossification systems have only been reported for the assessment of peak height velocity [46] and skeletal maturity in AIS [41,47], and have not been incorporated into guidelines around scoliosis bracing. There is also published work that recognises that an upper limit for bracing of 40° may well be too conservative. An upper limit of 40° was based on the lower limit of 50°, used as the lower limit of the surgical range [35]. A recent meta-analysis has suggested that bracing in AIS beyond 40° in skeletally immature patients could alter the clinical course [48], which confirms the findings of La Maida et al. [49] and Zhu et al. [50]. Other factors have also been shown to be important when assessing the success of bracing, and this is perhaps most true when assessing curves that are greater than 40° in size. Aulisa et al. have shown that vertebral body rotation is also key in predicting success or failure in bracing over 40° [51], with failure of bracing and conversion to surgery also indicated by younger age, open tri-radiate cartilages and lesser brace correction [52].

The ability to access healthcare to receive treatment promptly is also an issue in maximising the effects of any bracing guidelines. Ethnicity and socioeconomic status are known to indicate a greater likelihood of a larger curve at first presentation, and consequent surgical management [53]. This was also seen in non-operative treatment, with those of Afro-Caribbean descent being more likely to present with curves outside the surgical range at first presentation [54]. The waiting list to access healthcare once referred in those who will undergo surgical correction of their scoliosis with increasing curve sizes, complexity and magnitude of surgery and poorer outcomes has been well described [55,56,57]. Whilst bracing is not surgery, it is highly likely that the same arguments will equally apply to those awaiting an appointment to initiate the bracing process, be it with a physician or orthotist.

There are limitations to this work. First, this is a retrospective review over different time periods from three centres in the UK that deal with AIS, and, consequently, the results may not be representative of the whole country. However, it includes a large number of individuals, reviewed in centres across different geographical areas, with populations of different ethnicities, which does mitigate against this somewhat. Second, whilst all of the SRS bracing criteria [14] were applied fully, the BSS criteria [15] did not have the anatomical level of the apex of the curve (T7 or below) applied, as this information was not available. Thus, the results described using the BSS bracing criteria [15] must be viewed as a ‘best case scenario’, given that some of the individuals deemed suitable for bracing in the analysis may have been unsuitable, had the anatomical level of the curve apex been known. Third and finally, there are a number of reasons why individuals choose to undergo surgery. It is acknowledged that the decision to perform corrective surgery in AIS is complex and must consider factors other than the coronal Cobb angle alone. However, given the correlation between curve magnitude and quality of life, curve size measured using the Cobb angle remains a valuable indicator [58].

## 5. Conclusions

In conclusion, the data presented here demonstrate that only a small number of adolescents presenting with AIS to tertiary scoliosis centres in the UK would have met the criteria to be suitable for management with a brace. Whilst scoliosis surgery will remain, it is the authors’ view that, were the delays to review by a specialist reduced or eliminated, and the acceptable parameters within the criteria for bracing altered to be more inclusive, a greater number of adolescents would potentially benefit from a spinal brace, which could reduce the need for surgical intervention.

## Figures and Tables

**Figure 1 healthcare-11-00445-f001:**
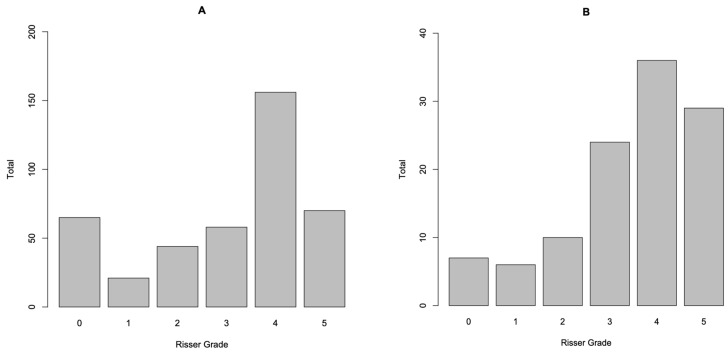
(**A**,**B**) The number of individuals for each Risser grade at presentation for both females (**A**) and males (**B**).

**Figure 2 healthcare-11-00445-f002:**
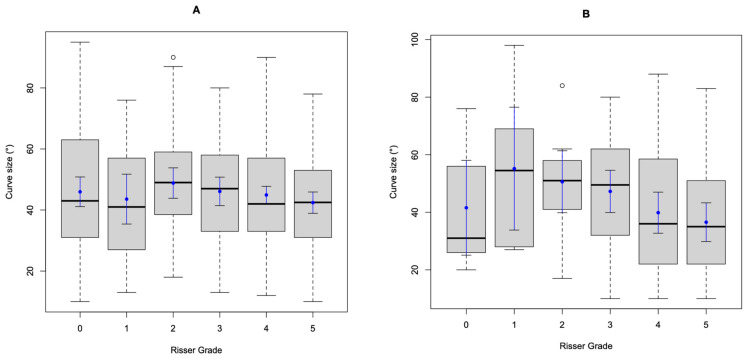
(**A**,**B**) The size of the curve stratified by Risser grade for females (**A**) and males (**B**) as box and whisker plots. The heavy line in the box is the median value, with the box representing the interquartile range (IQR). The whiskers represent 1.5 times the IQR, and all data outside of the whisker are represented as an open circle. The solid blue circle is the mean, and the bars from that circle are the 95% confidence intervals of the mean.

**Table 1 healthcare-11-00445-t001:** The demographics of the cohort.

	Female	Male	Statistical Significance between Females and Males
Sex	414 (79% of total cohort)	112 (21% of total cohort)	
Age (years)—mean, standard deviation and range	14 (SD 1.9, range 10–18)	15 (SD 1.6, range 11–18)	*p* < 0.001
Main thoracic curves (Lenke 1)	315 (76% of all females)	95 (85% of all males)	*p* = 0.064
Double major curves (Lenke 3)	92 (22% of all females)	11 (10% of all males)	*p* = 0.005
Thoracolumbar curves (Lenke 5)	7 (2% of all females)	6 (5% of all males)	*p* = 0.061

**Table 2 healthcare-11-00445-t002:** The number of individuals suitable for bracing, as per the Scoliosis Research Society (SRS) and British Scoliosis Society (BSS) criteria [14,15].

	Female	Male	Statistical Significance between Females and Males
SRS—curve 20–40°, Risser 0–1	35 (8% of all females)	6 (5% of all males)	*p* = 0.376
SRS—curve 30–40°, Risser 2–3	18 (4% of all females)	3 (3% of all males)	*p* = 0.600
BSS—age 10–15 years, curve 20–40°, Risser 0–2	43 (10% of all females)	7 (6% of all males)	*p* = 0.253

## Data Availability

The data reported in this study are available from the corresponding author following reasonable request.

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
