# Peer review of "An Examination of the Number of Adolescent Scoliotic Curves That Are Braceable at First Presentation to a Scoliosis Service"

_healthcare, 2023, doi:10.3390/healthcare11030445_

Round 1

Reviewer 1 Report

The objective of the study is very much relevant and important for the treating surgeons. Management of Adolescent Idiopathic Scoliosis with brace has lots of advantages over surgery which are discussed in detail in the study. This study has shown that although many AIS cases can be managed with brace why very few cases are able to reach to the tertiary center at the stage of bracing.

This is a well written article. Few comments from the reviewer are mentioned below

Comment on title:

The title of the article is confusing. The question “Is bracing useful”; followed by another question is not that clear. Looking at objectives of the study it seems that author wants to show the number/percentage of patients who present first time at the tertiary referral center who can be managed by bracing based on SRS or BSS criteria. This is not reflected in the title. The study population is adolescent idiopathic scoliosis so it is better to mention Adolescent Idiopathic Scoliosis in the title.  

Comments on result section:

In the result section (line 113 and 114), it is better to mention 42 cases of early onset idiopathic scoliosis which includes Infantile idiopathic scoliosis and juvenile idiopathic scoliosis because early onset scoliosis includes congenital, neuromuscular or syndromic as well.

The significance of the correlation between curve magnitude and risser’s stage is not clear although it is mentioned in the result.

It is shown from the study that only 10% of the cases presenting in the tertiary care center can be managed with brace according to SRS or BSS criteria. The first issue that can be thought of is based on these criterias very few are numbers can be managed with brace that means do we need to look upon the reliability of these criteria, do we need to revise it and probably this is not the issue in this study. The second issue is why very few numbers of cases are presenting in these centers although bracing is effective treatment method according to BRAIST trial. So, it would have been better if the causes of delay would have been studied and mentioned in results.

Comments of discussion section:

Causes of delay are mentioned in the discussion section but it would have been even better if they were discussed correlating with the findings from the study regarding the delay in presentation of tertiary care center.

Author Response

Thank you for the review.

The title has been changed to clarify the research that is being presented

We have not changed the section relating to the number of exclusions secondary to other diagnoses. This is because all children were older than 10 years and thus are not defined as Early Onset Scoliosis. Whilst we accept that they were likely to have had their scoliosis when under 10 years of age, we cannot confirm that with 100% certainty.

With regards to the comment around the correlation between curve size and risser status, no correlation is claimed or calculated. Both terms are documented as results as they both form independent parts of the bracing criteria and are documented as such. However, in expanding the discussion, we have considered at length the issues of inaccuracy of the Risser status in the assessment of peak height velocity and assessed the pros and cons of other systems of ossification for this purpose.

As for the appropriateness of the bracing criteria, this is now discussed fully in the discussion section, in terms of both skeletal maturity assessment and the size of curve appropriate for bracing.

Unfortunately, no comment can be made on the individual reasons for delay to presentation as this information is not available. Thus, as is in the current discussion, only general comments can be made.

Reviewer 2 Report

Dear author, I read your paper with great interest. I congratulate you on the originality of this study. The presentation is clear as well as the methodology and the result of your study very interesting but also surprizing.

However, I have few comments and questions:

First, I consider that the title (which is otherwise too long) does not really reflect the content and conclusions of your paper.

The question is not whether treatment with a brace is useful (that is already well documented in the literature) but whether appropriate screening would improve management of AIS and reduce the number of patients eligible for surgery.

So, I will suggest this title. "Screening of Adolescent idiopathic scoliosis in UK: How to brace adequately and how to avoid surgery?"

Line 29: Etiology is not unknown but unclear and multifactorial. Ad some references concerning pathogeny

Line 32: indication for surgery it is not only the Cobb angle but also frontal and the sagittal plane balance are another criterion for surgery

Your study contains three population groups with significant disparities which can be a bias for your study.

In fact, we know that socio-economic level of a population can influences on the time to diagnosis. This could be reflected in your study.

Furthermore, the screening of scoliosis by the general practitioner is based on what clinical criteria?

Based on the results of your study, what would you recommend improving the time between initial diagnosis and referral to a tertiary referral centre? This could be included in your conclusions.

Author Response

Thank you for the in depth review.

We have altered the title to better reflect the research reported here. We have changed it in line with another reviewer. We hope that the change in title meets with your approval.

We have added text to clarify that the indications for surgery include the size of the curve, the amount of rotation and the sagittal balance as you suggest.

Whilst we are unable to make any comment over the actual socioeconomic status for the individuals in this study as that information is not available for review, we have made reference to the general population in each of the three catchment areas and have also made reference the effects of socioeconomic class and delays in treatment in our expanded discussion.

We are not able to say what the screening criteria used in primary care are for the individuals in this study as this information does not exist. Our experience suggests that this is only by clinical examination and does not follow a structured format. We would agree however that reducing the time from primary care referral to assessment by a scoliosis expert would be appropriate and have added text to this effect. In the expanded discussion we now also discuss the pros and cons of school screening.

Reviewer 3 Report

Comments

The paper is an excellent contribution in the field of spinal scoliosis and is publishable. For me, the following points must be considered before publication.

 1.      Title: Title is too complex and can be shortened for better understanding of the reader. The title also does not describe the comparative analysis carried out between male and female cases.

2.      Abstract: The abbreviations of the society must appear in full form when used for the first time.

3. Figure 1: is not completely displayed and need consistency with alphabetic representation (uppercase OR lowercase letters)

4.      Discussion: Discussion is too general and lacks points explaining the results of the study reading the comparative analysis between male and female. The discussion should be modified so the results of this study could be better comprehend by the reader e.g. differences observed in the tables and figures regarding gender comparison, their significance and any possible reason for more female cases presented.

Author Response

Thank you for your review. The title has been altered to better reflect the research presented.

The society abbreviations have been addressed.

We can confirm that Figure 1 is appropriately labelled but we are aware that the figure was altered in the submission process previously (by the software). We apologise for this but it is out of our hands. A PDF of the figure was also submitted as per the instructions to authors.

The number of females outweighs the number of males and this fits with the known presentation of AIS. Text is added to this effect.